# The effects of rifaximin and lactulose on the gut-liver-brain axis in rats with minimal hepatic encephalopathy

Xueyan Lin[1,2,3☉], Zhengchao Zhang[4,5,6☉], Yi Lin[1,2,3], Shiyun Lu[1,2,3*], Rongrong Chen ![ORCID][1,2,3*]

1 Department of Gastroenterology and Hepatology, The Shengli Clinical Medical College, Fujian Medical University, Fuzhou, China, 2 Department of Gastroenterology and Hepatology, Fujian Provincial Hospital, Fuzhou, China, 3 Department of Gastroenterology and Hepatology, Fuzhou University Affiliated Provincial Hospital, Fuzhou, China, 4 Department of Emergency surgery, The Shengli Clinical Medical College, Fujian Medical University, Fuzhou, Fujian, China, 5 Department of Emergency surgery, Fujian Provincial Hospital, Fuzhou, Fujian, China, 6 Department of Emergency surgery, Fuzhou University Affiliated Provincial Hospital, Fuzhou, China

☉ These authors contribute equally.
* lushiyun121739@163.com (SL); 18650303056@163.com (RC)

## Abstract

### Background

The principal therapeutic agents for minimal hepatic encephalopathy (MHE), which focus on the modulation of the gut microbiota, include lactulose and rifaximin; however, the precise mechanisms through which they operate remain unclear.

### Aim

This study aimed to investigate the effects of rifaximin and lactulose on the gut-liver-brain axis in a rat model of MHE and to clarify the underlying mechanisms involved.

### Methods

A rat model of MHE was established by subcutaneous carbon tetrachloride (CCl4) injection. The Morris water maze (MWM) test was used to assess cognitive function in MHE rats following treatment with rifaximin and lactulose. Serum and cerebrospinal fluid ammonia levels were quantified, along with measurements of portal lipopolysaccharide (LPS) and various serum inflammatory markers. Furthermore, the expression of Toll-like receptor 4 (TLR4) in the liver was examined by histopathological evaluation. Additional analyses included the detection of tight junction proteins in the intestinal mucosa as well as colon fecal 16S rRNA sequencing and metabolic pathway assessments.

**Data availability statement:** All relevant data are within the manuscript and its Supporting Information files.

**Funding:** We confirm that this study received funding support from the following sources: • Fujian Medical University QiHang fund (2019QH1167 and 2019QH1185) • Fujian Provincial Health Technology Project (2022QNA003) • Science and Technology Planning Project of Fujian Provincial Health Commission (2020GGA002) • Fujian Provincial Natural Science Foundation (2020J05264 and 2023J011210) However, we would like to clarify that these grants were used solely to support research-related expenses and did not result in any financial compensation or personal remuneration for the authors. Therefore, no financial disclosure is applicable.

**Competing interests:** The authors have declared that no competing interests exist.

## Results

Both rifaximin and lactulose were effective in reducing ammonia concentrations in MHE rats and ameliorating cognitive deficits, although they exhibited a minimal impact on hepatic function. Post-treatment assessments revealed significant reductions in portal LPS, serum interleukin-1β (IL-1β), and tumor necrosis factor-α (TNF-α). The expression of TLR4 in the liver and hepatic inflammatory infiltration were notably diminished. Rifaximin administration led to increased occludin expression in the intestinal tissues of MHE rats. Despite no significant alterations in the diversity or composition of the gut microbiota, metabolic pathway analyses indicated a downregulation of glycometabolism pathways following treatment.

## Conclusion

Rifaximin and lactulose may enhance cognitive performance in MHE rats by modulating gut microbiota metabolism and preserving the intestinal barrier integrity. This modulation is associated with lowered ammonia levels, decreased translocation of LPS to the liver, and reduced inflammatory response, both in the liver and systemically.

## 1 Introduction

Hepatic encephalopathy (HE) encompasses a range of neuropsychiatric disorders arising from metabolic disturbances associated with acute or chronic liver dysfunction and aberrant portal-systemic circulation. The onset is often a prognostic indicator of poor outcomes in patients with liver disease [1]. Minimal hepatic encephalopathy (MHE), an early and typically asymptomatic stage of HE, is characterized by subtle cognitive impairments detectable only through specialized neuropsychological and neurophysiological assessments [1]. Recent surveys have highlighted the lack of awareness and testing for HE among medical professionals. A 2020 study in Germany found that only 26.7% of gastroenterologists routinely test cirrhotic patients for HE, whereas 53.7% of general practitioners were unaware of MHE [2]. Similar trends were observed in India, where 63% of doctors reported never testing for MHE [3]. The prevalence of MHE ranges from 20% to 80% due to its subtle symptoms and varying diagnostic criteria [4]. A survey of 16 hospitals in 13 regions of China showed an MHE incidence of 39.9% [5], while in the U.S., rates reached as high as 60%−80% [6]. Although not fatal, MHE affects patients' quality of life, sleep, and driving ability, imposing significant economic and social burdens [7,8]. Lack of awareness may delay treatment and lead to disease progression.

Prior research has established that the pathogenesis of MHE bears similarities to that of HE; however, the exact mechanisms remain inadequately characterized. Current consensus suggests that the accumulation of ammonia originating from the gut and entering the central nervous system, is the primary precipitating factor for MHE [9–12]. Ammonia is absorbed from various sites, including the intestine, and

enters the portal circulation of the liver. Under normal liver function, ammonia is converted into urea via the urea cycle, and is then excreted by the kidneys [13].Numerous studies have proposed that the pathophysiology of hepatic encephalopathy may be intertwined with the activity of gut microbiota. Dysbiosis within the gut microbiome exacerbates systemic inflammation, which can lead to neuroinflammation and subsequent neurological dysfunction, including HE [14–16]. Furthermore, emerging studies have demonstrated that bacterial metabolites significantly influence the gut-liver-brain axis. In patients with cirrhosis and MHE, a decline in secondary bile acid formation, alongside alterations in iso- and oxo-bile acids, has been documented [17].

Although lactulose and rifaximin have demonstrated clinical efficacy in alleviating symptoms of MHE, their precise mechanisms of action-particularly how they influence the gut-liver-brain axis-remain incompletely understood [18–20]. Conflicting findings exist regarding whether these agents truly modulate the gut microbiota, with some studies reporting negligible changes in microbial diversity or abundance despite clinical improvements [21–23]. This discrepancy raises important unresolved questions: Do these treatments exert their therapeutic effects via alterations in microbial metabolites, inflammatory signaling, or bile acid profiles rather than through broad microbial compositional shifts? What are the key molecular mediators linking microbiota alterations to neurological function in MHE? Addressing these questions is clinically significant because current treatment strategies lack mechanistic precision, limiting the development of targeted or adjunctive therapies. In particular, understanding the interplay among microbial metabolites, hepatic metabolism, and neuroinflammation could offer insights into more personalized and effective approaches for MHE management.

Therefore, this study aimed to investigate the functional effects of lactulose and rifaximin on the gut-liver-brain axis using a rat model of MHE. By assessing cognitive performance, inflammatory markers, ammonia levels, bile acid composition, and gut microbiota-derived metabolites, we sought to elucidate the specific pathways through which these agents exert their effects. This research builds upon prior clinical and animal studies by integrating behavioral, biochemical, and microbial assessments, thereby addressing a critical gap in the mechanistic understanding of MHE therapies.

## 2 Materials and methods

### 2.1 Animal

A total of 26 male Sprague-Dawley rats, each weighing 160-180g and classified as Specific Pathogen-Free (SPF), were sourced from Beijing Huafukang Biotechnology Co., Ltd. (License No. SCXK [Beijing, 2019−0008]). All experimental procedures were conducted in accordance with the 3R principles for the use of experimental animals and were approved by the Animal Ethics Committee of the School of Basic Medicine, Fujian Medical University (Document No.: 2021101201). The rats were housed in cages under controlled environmental conditions, including a 12-hour light/dark cycle, a humidity range between 40% and 70%, and a temperature maintained at 20 °C to 26 °C. They were provided with ad libitum access to food and water. All experimental procedures adhered to the ethical standards set forth in the "Guide for the Care and Use of Laboratory Animals" as outlined by the National Academy of Sciences and published by the National Institutes of Health.

### 2.2 Experimental design

An established MHE model was used as described in previous studies [24]. After a one-week acclimatization period, 26 rats were stratified by weight and randomly assigned to a control group (n = 6) or an MHE model group (n = 20). The MHE group received subcutaneous injections of CCl4 (3 ml/kg, 1:1 with olive oil; Sinopharm Chemical Reagent Co.) twice weekly, along with 5% ethanol in their drinking water and a standard diet for nine weeks. The control group received saline injections and had unrestricted access to food and water. After nine weeks, brainstem auditory evoked potential (BAEP) tests confirmed MHE [21], with two rats excluded based on abnormal results (Supplementary Table 1). The remaining MHE rats underwent cognitive assessment using the Morris water maze (MWM) test and were divided into groups: normal

saline (MNS), lactulose (ML), and rifaximin (MR), with the control group as normal controls (C), each with 6 rats. At the end of the 9th week, C and MNS groups were gavaged with saline, while ML and MR groups received lactulose (667 g/L lactulose, Drug Industries Company Abbott, Egypt) at a dose of 10 mL/kg body weight and rifaximin (Alfasigma S.p.a. Co., Ltd.) at a dose of 50 mg/kg body weight respectively, once daily for 8 weeks (Fig 1). During the experiment, all rats had free access to standard food and water. At the end of the experiment, all rats were euthanized using a carbon dioxide ($CO_2$) chamber to ensure compliance with animal ethical guidelines and minimize animal suffering to the greatest extent possible.

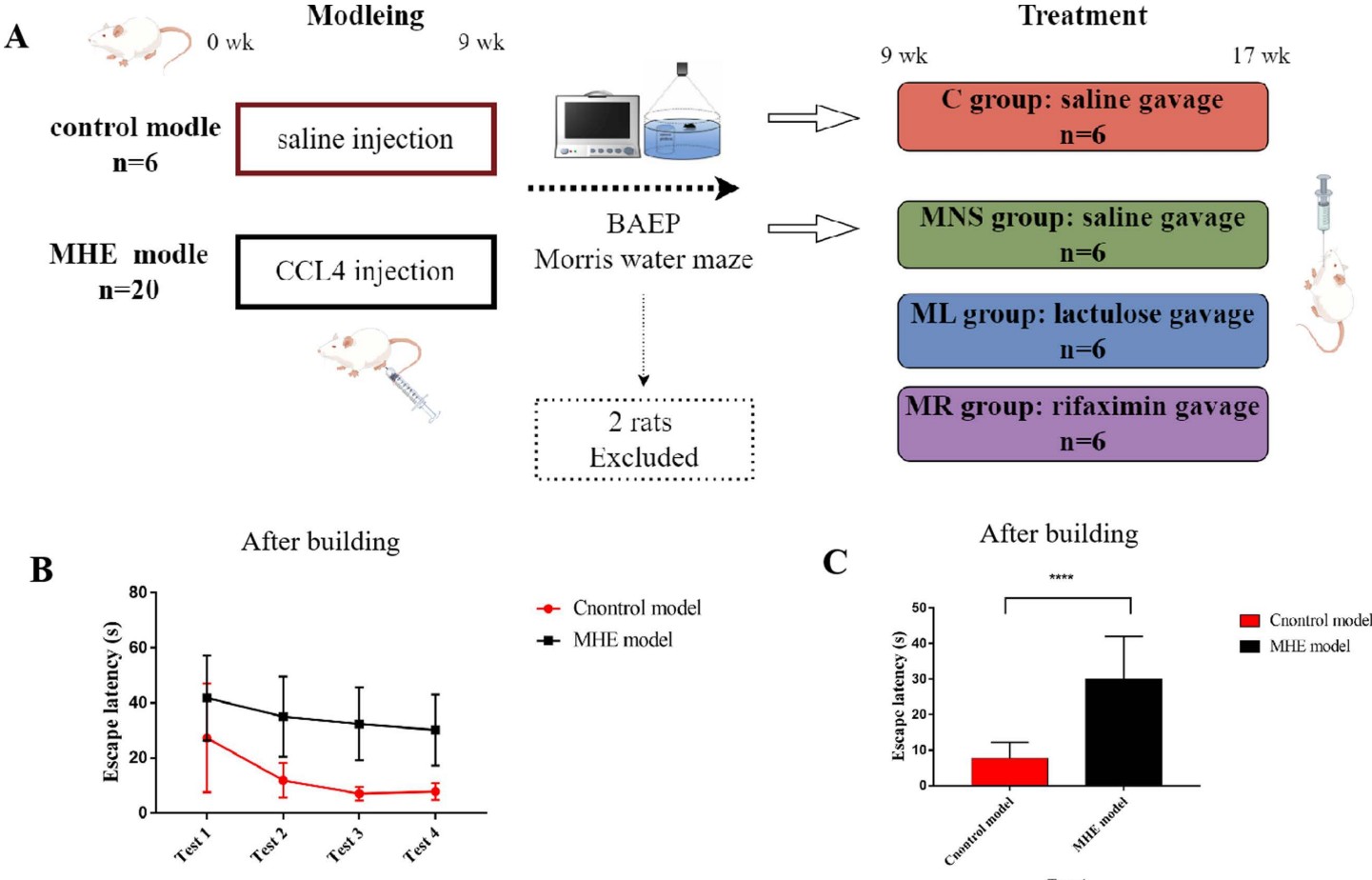

**Fig 1. Experimental design. (A)** 26 rats were randomly divided into a control model (n=6) and MHE model (n=20). The MHE model rats were given subcutaneous injections of 3 mL/kg of CCl4 solution (a mixture of $CCl_4$ and olive oil at a ratio of 1:1 v/v) twice a week and were fed 5% alcohol in drinking water for 9 consecutive weeks. Control model rats received equal volume normal saline subcutaneous injections twice a week and fed regular water. 9 weeks after injection, BAEP of rats should be tested to confirm the diagnosis of MHE through latency of BAEP and Morris water maze test. Then, rats with MHE and cognitive impairment were screened and randomly divided into normal saline group (MNS group), lactulose, group (ML group), and rifaximin group (MR group), and the control model rats into normal control group (C group), with 6 rats in each group. The C and MNS group were gavaged with equal volume normal saline, the ML and MR group gavaged with lactulose at a dose of 10 mL/kg body weight and rifaximin at a dose of 50 mg/kg body weight respectively once a day for 8 weeks until the end of the experiment. **(B)** The escape latencies of four test trials of the Morris water maze in 2 models. The escape latencies of control model gradually decreased as training time increased. Compared to controls, MHE model had significantly longer escape latencies during the 2nd, 3rd and **(C)** 4th tests ($p<0.001$). Abbreviations: MHE, minimal hepatic encephalopathy; $CCL_4$, carbon tetrachloride; BAEP, brainstem auditory evoked potentials; Results are shown as the means ± SD. **** $p<0.0001$, as determined by Mann-Whitney U test between two groups.

## 2.3 Morris water maze

MWM is generally accepted to assess spatial learning ability in rats [25,26], both modeling and post-treatment. The setup consisted of a cylindrical water tank, measuring 150 cm in diameter and 60 cm in height, filled with water maintained at a temperature of 25 ± 1°C, rendered opaque with black ink. The tank was sectioned into four quadrants, with a platform (10 cm in diameter) submerged 2 cm below the water surface positioned at the center of one quadrant. The MWM assessment was conducted each morning over four consecutive days. Rats were placed at a predetermined starting location within the pool and allowed to swim freely until locating the platform, with the latency to escape recorded for analysis.

## 2.4 Enzyme-linked immunosorbent assay (ELISA) analysis of circulating pro-inflammatory mediators

To detect IL-1β, TNF-α, and LPS, serum samples were collected from each group via portal and tail veins. These samples were then analyzed using a Rat IL-1β ELISA kit (MM-0047R1, Jiangsu Enzyme Immunoassay Industry Co., Ltd.), a Rat TNF-α ELISA kit (MM-0180R1, Jiangsu Enzyme Immunoassay Industry Co., Ltd.), and a Rat LPS ELISA kit (MM-20584R1, Jiangsu Enzyme Immunoassay Industry Co., Ltd.). For the determination of toll-like receptor 4 (TLR4) protein, liver tissue homogenates were analyzed using a Rat TLR4 ELISA kit (MM-0221R1, Jiangsu Enzyme Immunoassay Industry Co., Ltd.). All samples were processed and assayed following the manufacturer's protocols.

## 2.5 Liver functional tests and ammonia level detection

The serum liver function in the rats were measured using alanine aminotransferase (ALT) Assay kit (70111, Shandong Boke Biotechnology), aspartate aminotransferase (AST) Assay kit (70110, Shandong Boke Biotechnology), albumin (ALB) Test kit (70113, Shandong Boke Biotechnology), and total bilirubin (TBIL) Test kit (70182, Shandong Boke Biotechnology), respectively. The concentrations of ammonia in serum and cerebrospinal were measured using the Ammonia Assay kit (E-BC-K145-M, Elabscience), according to the manufacturer's protocol.

## 2.6 Hematoxylin and eosin staining

Fresh liver tissues were preserved in 4% paraformaldehyde, subjected to dehydration through a series of graded ethanol solutions, embedded in paraffin, and then sectioned. The slides were baked and subsequently stained using hematoxylin and eosin.

## 2.7 WB detection

Take a certain amount of rat colon tissue, add RIPA lysis solution, grind with a tissue grinder, and extract total tissue protein. Centrifuge at 4 °C for 10 minutes using a 12000 r/min high-speed centrifuge. Take the supernatant and use the BCA Protein Assay Kit (E-BC-K318-M, Elabscience) to quantify the total protein. After denaturing the protein sample, conduct sodium dodecyl sulfate gel electrophoresis (SDS-PAGE) for 1.5h, and then use 300mA constant flow membrane for 1.0-2.0h. After sealing the PVDF membrane with skim milk powder, the rabbit anti occludin antibody (DF7504, Affinity, 1:1000) or Mouse Monoclonal Anti-β ACTIN (TA-09, ZSGB, 1:5000), was incubated overnight at 4 °C. The next day, the membrane was treated with a secondary antibody on a shaker at room temperature for 1 hour. After the incubation, the membrane was immersed in a chemiluminescent solution and developed using a high-sensitivity chemiluminescence imaging system.

## 2.8 Analysis of intestinal flora by 16SrRNA sequencing

Total bacterial DNA was extracted from a 200 mg colonic content sample using the E.Z.N.A.™ Mag-Bind Soil DNA Kit (M5635-02, OMEGA) following the manufacturer's protocol. DNA concentration was measured with a Qubit 4.0 (Thermo, USA) to ensure sufficient yield of high-quality DNA. The V3–V4 region of the bacterial 16S rRNA gene was targeted for

PCR amplification using 2×Hieff® Robust PCR Master Mix (Yeasen, 10105ES03, China). Universal PCR primers (forward: CCTACGGGNGGCWGCAG, reverse: GACTACHVGGGTATCTAATCC), purified via PAGE, were used. PCR products were checked on 2% agarose gels in TBE buffer and visualized with ethidium bromide under ultraviolet light. Purification to remove excess primers was done using Hieff NGS™ DNA Selection Beads (Yeasen, 10105ES03, China). Sequencing was performed on the Illumina MiSeq platform (Illumina MiSeq, USA), and the resulting short reads were assembled into longer sequences using PEAR software (version 0.9.8). These sequences were then processed and analyzed with standard bioinformatic methods.

## 2.9 Statistical analysis

Statistical analyses were performed using SPSS 25.0 (IBM Corp., Armonk, NY, United States). The statistical analyses and graphing were conducted using GraphPad Prism 7.0 (GraphPad Software, La Jolla, CA, USA). Data are presented as means±SD of at least three independent experiments. Group comparisons were performed using the Mann-Whitney U test. One-way ANOVA was utilized for the comparison of multiple groups, followed by the Student-Newman-Keuls post-hoc test when deemed necessary. A $p$-value of less significance was defined as $p$-value$<0.05$.

## 3 Experimental results

### 3.1 Effects of rifaximin and lactulose on BAEP, cognitive impairment, ammonia levels, and hepatic function in MHE rats

The administration of rifaximin and lactulose resulted in a significant reduction in the latency of BAEP I and escape latency, indicating an enhancement of cognitive function in MHE rats (Fig 2A and 2B). In the normal control group, the average serum ammonia level was recorded at 0.20 mmol/L, while MHE rats exhibited an increase to 0.45 mmol/L. Following treatment with rifaximin and lactulose, serum ammonia levels decreased to 0.22 mmol/L and 0.31 mmol/L, respectively, demonstrating statistical significance (as shown in Fig 2C). Comparable effects were also apparent in cerebrospinal fluid ammonia concentrations (Fig 2D), suggesting that rifaximin and lactulose were effective in reducing circulating ammonia and subsequently ameliorating clinical symptoms. Additionally, immunofluorescence analysis revealed significantly increased Gamma-aminobutyric acid receptor-associated protein (GABARAP) expression in the brain tissue of MNS rats compared to controls ($p<0.05$). Notably, GABARAP levels were partially reduced following lactulose or rifaximin treatment (S5 Fig).

In the MNS group, elevated serum levels of ALT and AST were observed, alongside reduced ALB levels (S1A–S1C Fig). Nevertheless, no significant difference in TBIL was noted between the MNS and normal control groups (S1D Fig). While rifaximin treatment improved hypoalbuminemia, it did not significantly inhibit the rise in serum ALT, AST, or TBIL levels in MHE rats (S1 Fig).

### 3.2 Rifaximin and lactulose effects on LPS/TLR4 signaling in MHE rats

In MHE rats, serum LPS levels in the portal vein were markedly elevated (Fig 3A), corroborating the increased serum LPS concentration. In addition, circulatory pro-inflammatory mediators, including IL-1β and TNF-α, were heightened in the MNS group. However, these inflammatory markers were significantly diminished in the groups treated with rifaximin and lactulose (Fig 3A, 3B, and 3C).

To explore the effects of rifaximin and lactulose on LPS-induced inflammatory signaling in the liver of MHE rats, it was noted that subcutaneous CCl4 injection escalated LPS levels in the portal vein, instigating a host immune response to endotoxins. This subsequently resulted in a notable upregulation of TLR4 expression in the liver, which is crucial for LPS detection and signaling initiation (Fig 3D). Rifaximin and lactulose treatment successfully mitigated the rise in portal LPS levels, the upregulation of inflammatory mediators, and TLR4 expression in the liver, indicating a reduction in hepatic LPS translocation.

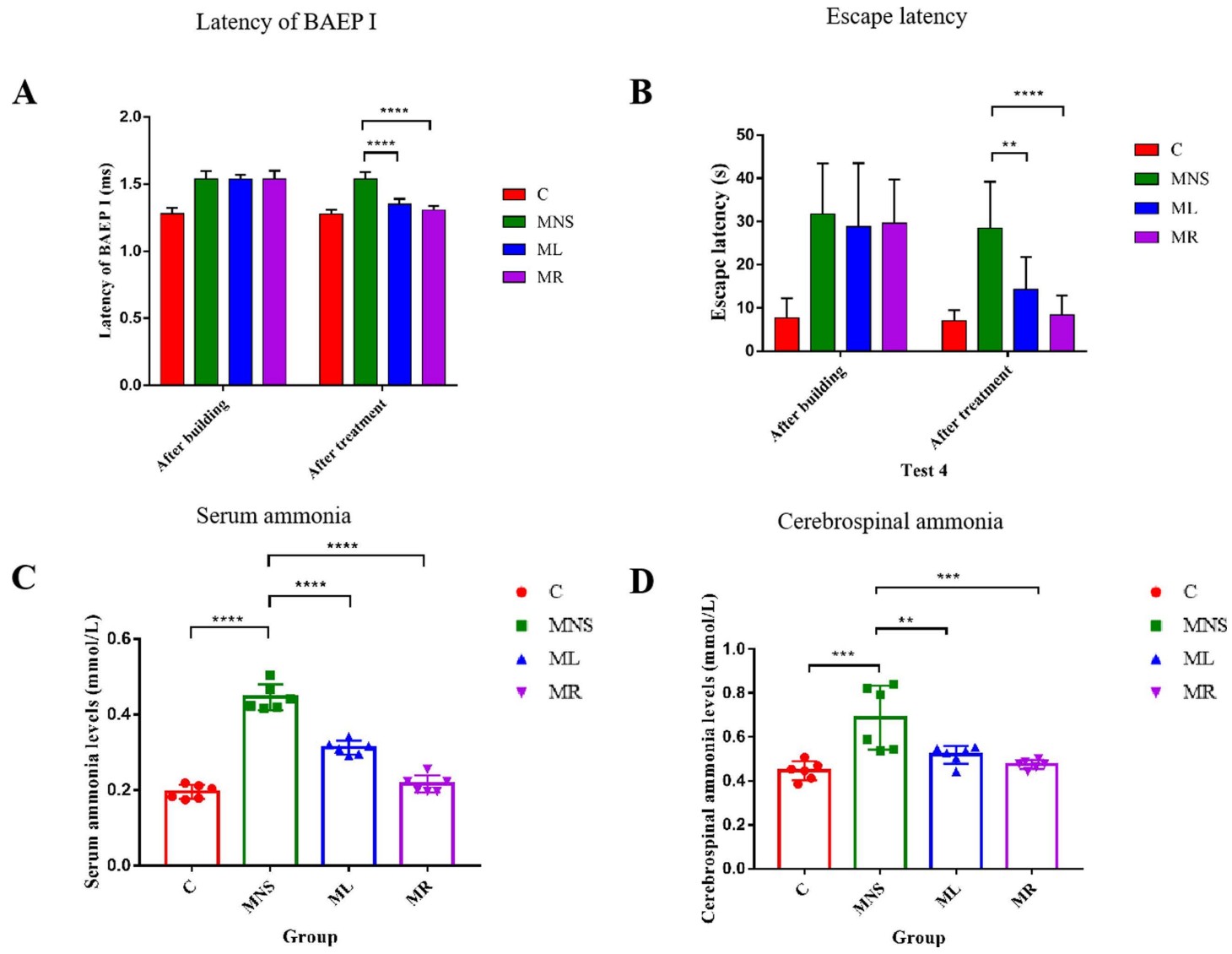

**Fig 2. BAEP, cognitive function, and ammonia levels in four groups. (A)** Latency of BAEP I and **(B)** Escape latency of the MNS group was significantly higher than that of the C group, then obviously decreased after the intervention of rifaximin and lactulose. **(C)** Serum ammonia and **(D)** Cerebrospinal ammonia levels decreased significantly upon the treatment of rifaximin and lactulose. Abbreviations: BAEP, brainstem auditory evoked potential; Results are shown as the means±SD. ** $p < 0.01$, *** $p < 0.001$, **** $p < 0.0001$, as determined by Mann-Whitney U test between two groups.

Histological evaluation through hematoxylin and eosin staining revealed hepatic inflammation in MHE rats, characterized by structural disruption of liver lobules, disordered liver cell arrangement, and significant infiltration of inflammatory cells. Importantly, treatment with rifaximin and lactulose considerably reduced hepatic inflammation (Fig 3E).

### 3.3 Effects of rifaximin and lactulose on the intestinal barrier in MHE rats

Both rifaximin and lactulose were found to diminish LPS translocation to the liver, as evidenced by the decreased expression of TLR4-mediated inflammatory mediators (Fig 3). To elucidate the mechanisms behind these effects, we assessed the integrity of intestinal tight junction proteins. Western blot analysis indicated a reduction in occludin expression in the

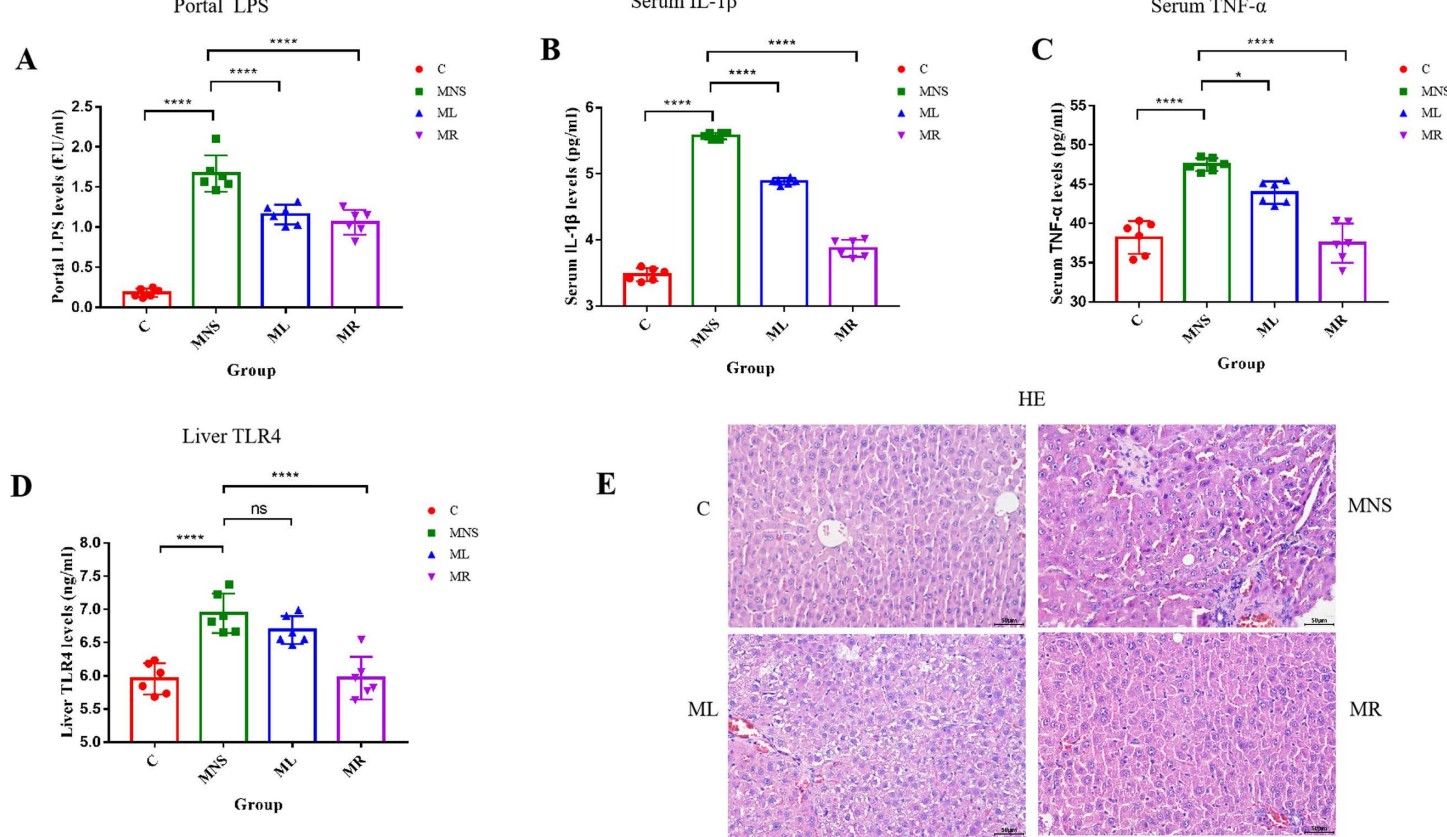

**Fig 3. Rifaximin and lactulose on inflammation in MHE rats.** **(A)** Serum LPS levels in the portal vein of the MNS group was significantly higher than that of the C group, then obviously decreased after the intervention of rifaximin and lactulose. Decreased **(B)** serum IL-1β and **(C)** Serum TNF-α were observed in the ML and MR groups. **(D)** The expression of TLR4 of the liver increased in MNS group, then significantly decreased after the intervention of rifaximin. While there was no significant difference after the treatment of lactulose. **(E)** HE staining of the liver showed intact liver lobular structure and normal liver cells in the C group. The MNS group showed structural destruction of liver lobules, disordered arrangement of liver cells, and significant inflammatory cell infiltration. The degree of liver damage is reduced after treatment with rifaximin and lactulose. Abbreviations: LPS, lipopolysaccharide; IL-1β, interleukin-1β; TNF-α, tumor necrosis factor-α; TLR4, toll like receptor 4; Results are shown as the means±SD. \*\*$p < 0.05$, \*\*\* $p < 0.001$, \*\*\*\* $p < 0.0001$, ns, no significant difference, as determined by Mann-Whitney U test between two groups.

intestines of MHE rats; however, rifaximin treatment resulted in a significant increase in occludin levels. Conversely, lactulose did not exhibit a comparable effect on occludin expression as observed with rifaximin (Fig 4).

To further evaluate intestinal barrier function, we also examined the expression levels of other key tight junction proteins Claudin-1 and ZO-1 in the small intestine. The results showed that rifaximin increased the expression of claudin and ZO-1 in small intestinal tissues (S4 Fig).

## 3.4 Effect of rifaximin and lactulose on the diversity and major compositions of gut microbiome

To investigate the impacts of rifaximin and lactulose on gut health, we performed a 16S rRNA sequencing of intestinal fecal samples.

**3.4.1 Impact of rifaximin and lactulose on gut microbiome diversity.** The Shannon and Simpson α diversity index showed no significant difference among C, MNS, ML, MR group (Shannon diversity index median [IQR] 4.06±0.57 at C vs 4.45±0.74 at MNS vs 4.44±0.02 at ML vs 4.53±1.76 at MR, $p = 0.77$; Simpson diversity index median [IQR] 0.07±0.07

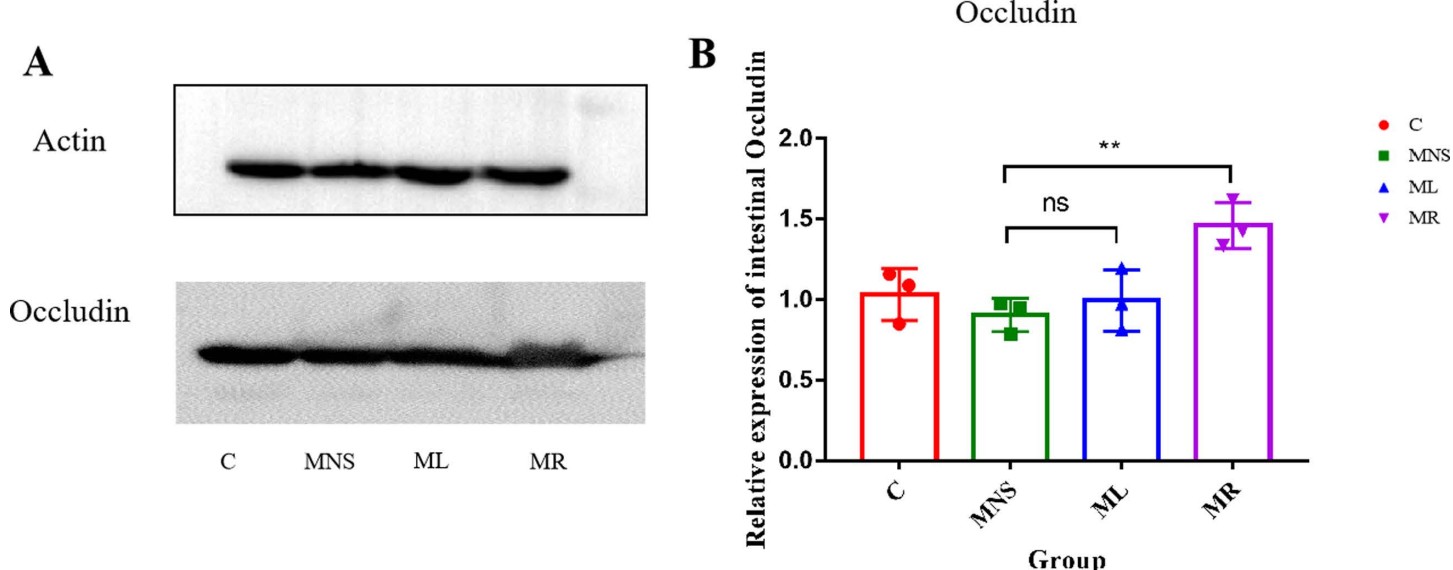

**Fig 4. The expression of tight junction proteins in intestinal wall. (A)** Western blots for occludin in the intestinal tissues. Actin was used as the loading control. **(B)** The relative expression of intestinal occludin analyzed by Image **J.** Abbreviations: Results are shown as the means ± SD. ** $p < 0.01$, ns, no significant difference, as determined by Mann-Whitney U test between two groups.

at C vs 0.03 ± 0.02 at MNS vs 0.03 ± 0.00 at ML vs 0.02 ± 0.21 at MR, P = 0.64) (Fig 5A). Principal co-ordinate analysis (PCoA) and principal component analysis (PCA) showed the samples of C group were clustered together, but MHE group (MNS group) samples induced remarkable changes in microbiome. The β diversity analysis showed no significant clustering differences in microbiota composition before and after treatment with rifaximin and lactulose (MR and ML groups) (Fig 5B).

**3.4.2 Influence of rifaximin and lactulose on the major compositions of the gut microbiome.** Fig 6 illustrates the changes in the composition of the gut microbiota among different groups. Both at the phylum and genus levels, a similar pattern of microbial community composition was found in the MR and the C group, which was obviously differed from that of the MNS and ML group. The gut microbiota composition after rifaximin intervention is closer to that of the normal control group, while the MHE group and the group after lactulose intervention had more similar gut microbiota compositions. The *Lactobacillus* abundance was highest in the normal control group and the rifaximin intervention group, and the *Allobaculum* abundance was higher in the MHE and lactulose intervention group. This suggests that rifaximin may be able to reverse the dominant bacterial genera in MHE (Fig 6A and 6B). The linear discriminant analysis effect size (LEfSe) was aligned with those of the analysis of the significant differences between groups. Then significant marker linear discriminant analysis (LDA) score obtained from LEfSe analysis. When LDA score was set to >4 in the LEfSe analysis, significantly different genera were identified among the four groups (Fig 6C). In addition, *Clostridiales* and *Spirochaetia* were enriched in the MNS group (LDA score >4) which were different with the C group. However, *Lactobacillus*, *Erysipelotrichia*, *Allobaculum*, and *Bifidobacterium* were enriched after treatment with rifaximin in the MR group. *Proteobacteria* and *Bacteroides* were enriched after treatment with lactulose in the ML group which show changes in microbial community structure different from the MR group. DESeq2 analysis was performed to compare differential genera among groups. The results showed the enrichment of 12 genera, and the lack of 6 genera (S2A Fig). Rifaximin treatment decreased the abundance of 5 genera and reduced the abundance of 8 genera (S2B Fig). The ML group decreased the abundance of 3 genera and reduced the abundance of 5 genera (S2C Fig). We found that *Saccharibacteria* was the most abundant

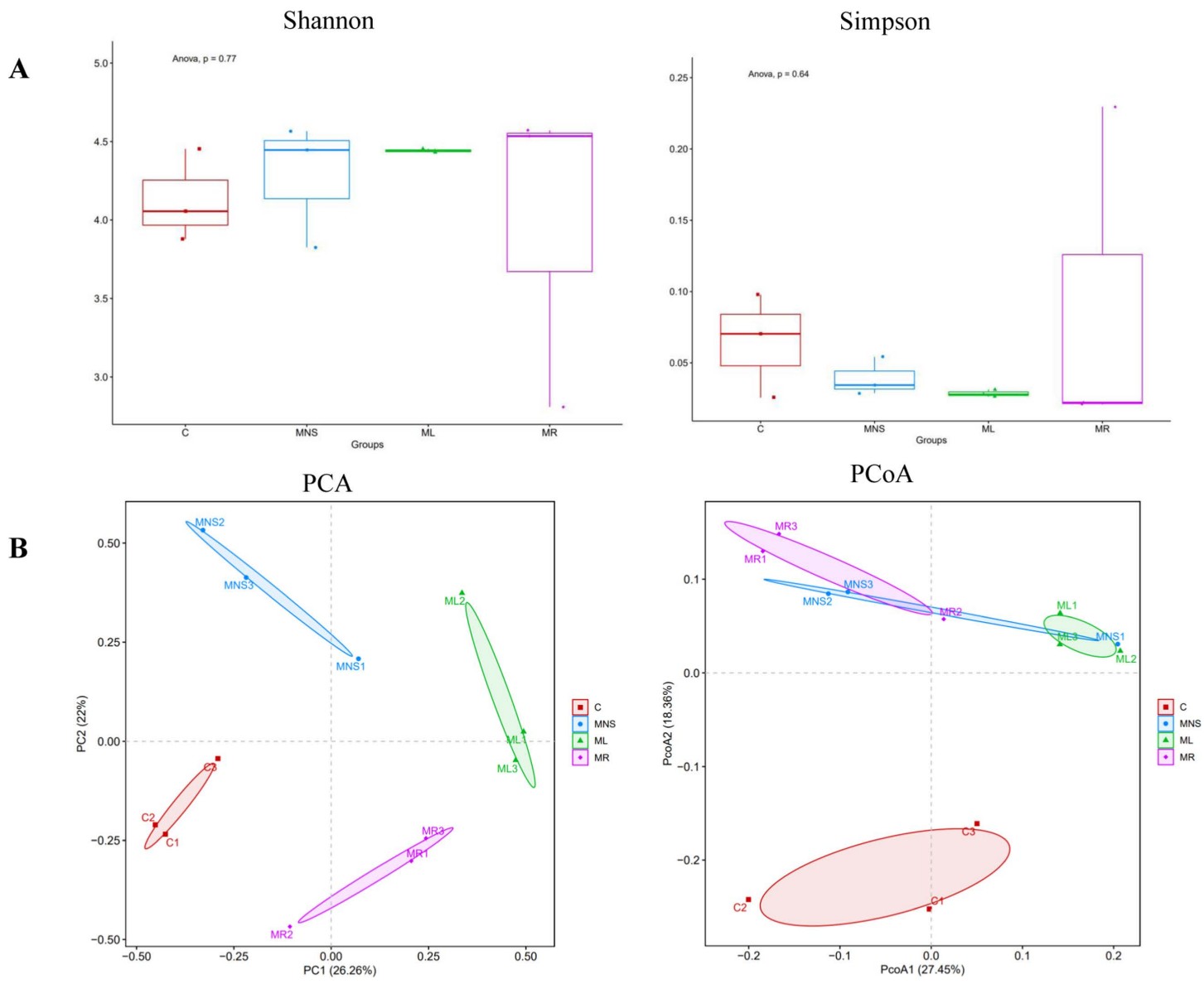

**Fig 5. Alpha and beta diversity of gut microbiome. (A)** The Shannon and Simpson α diversity index showed no significant difference among all groups (Shannon diversity index median [IQR] 4.06±0.57 at C vs 4.45±0.74 at MNS vs 4.44±0.02 at ML vs 4.53±1.76 at MR, $p$ = 0.77; Simpson diversity index median [IQR] 0.07±0.07 at C vs 0.03±0.02 at MNS vs 0.03±0.00 at ML vs 0.02±0.21 at MR, $p$=0.64). **(B)** Principal co-ordinates analysis (PCoA) and Principal component analysis (PCA) of gut microbiota, clear separation of samples by rifaximin and lactulose treatment was observed via PCoA and PCA. Abbreviations: Results are shown as the means±SD. $p < 0.05$, as determined by One-way ANOVA among four groups.

bacterial genus in the MNS group, and it significantly decreased after the treatment with rifaximin and lactulose. The *Lactobacillus* was the most abundant bacterial genus in the control group, and it significantly decreased in the MHE group. However, the abundance of *Lactobacillus* reversed after treatment with rifaximin and lactulose (S3 Fig).

### 3.4.3 Effect of rifaximin and lactulose on the metabolic features of gut microbiome.
Phylogenetic Investigation of Communities by Reconstruction of Unobserved States (PICRUSt) was employed to predict the metabolic pathways of the gut microbiota based on 16S rDNA gene sequencing. A comparison of the metabolic pathways revealed significant

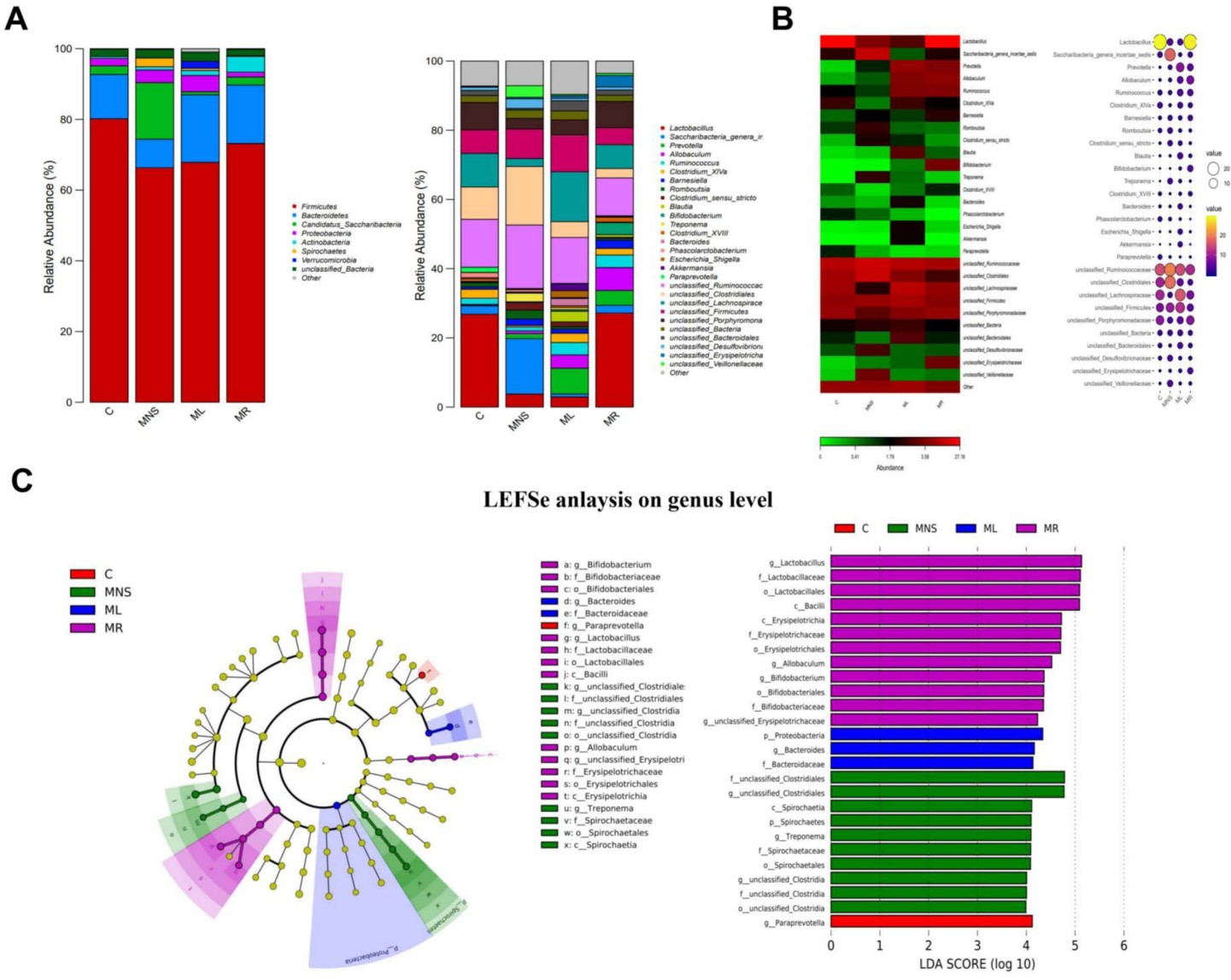

**Fig 6. Alterations in the composition of gut microbiome in phylum and genus. (A)** Alterations in the composition of gut microbiome in phylum and genus level. **(B)** Abundance heatmap and ballon map of genus. The darker the color (red) in the heatmap, the higher the abundance of the species, and the bluer the color, the lower the abundance of the species. The abundance information of the ballon map is represented by the size and color of the ballon. The yellower the color, the larger the ballon, and the higher the species abundance. **(C)** Differences in gut microbiota composition at the genus level were determined through LDA effect size analysis among the four groups (LDA score >4).

differences among the four groups (Fig 7A). The metabolic pathways obviously changed between the C and MNS group, downregulated pathways in the MNS group were glycometabolism (K02025 multiple sugar transport system permease protein, *p* = 0.0004; and K02026 multiple sugar transport system permease protein, *p* = 0.0012) (Fig 7B). Downregulated pathway after rifaximin treatment was K03406 methyl-accepting chemotaxis protein (*p* = 0.0099), the glycometabolism (K02025 multiple sugar transport system permease protein and K02026 multiple sugar transport system permease protein) and amino acid metabolism (K01834 phosphoglycerate mutase) were slight upregulated, but there were no significant differences (Fig 7C). After lactulose treatment, the pathways of glycometabolism (K02025 multiple sugar transport system

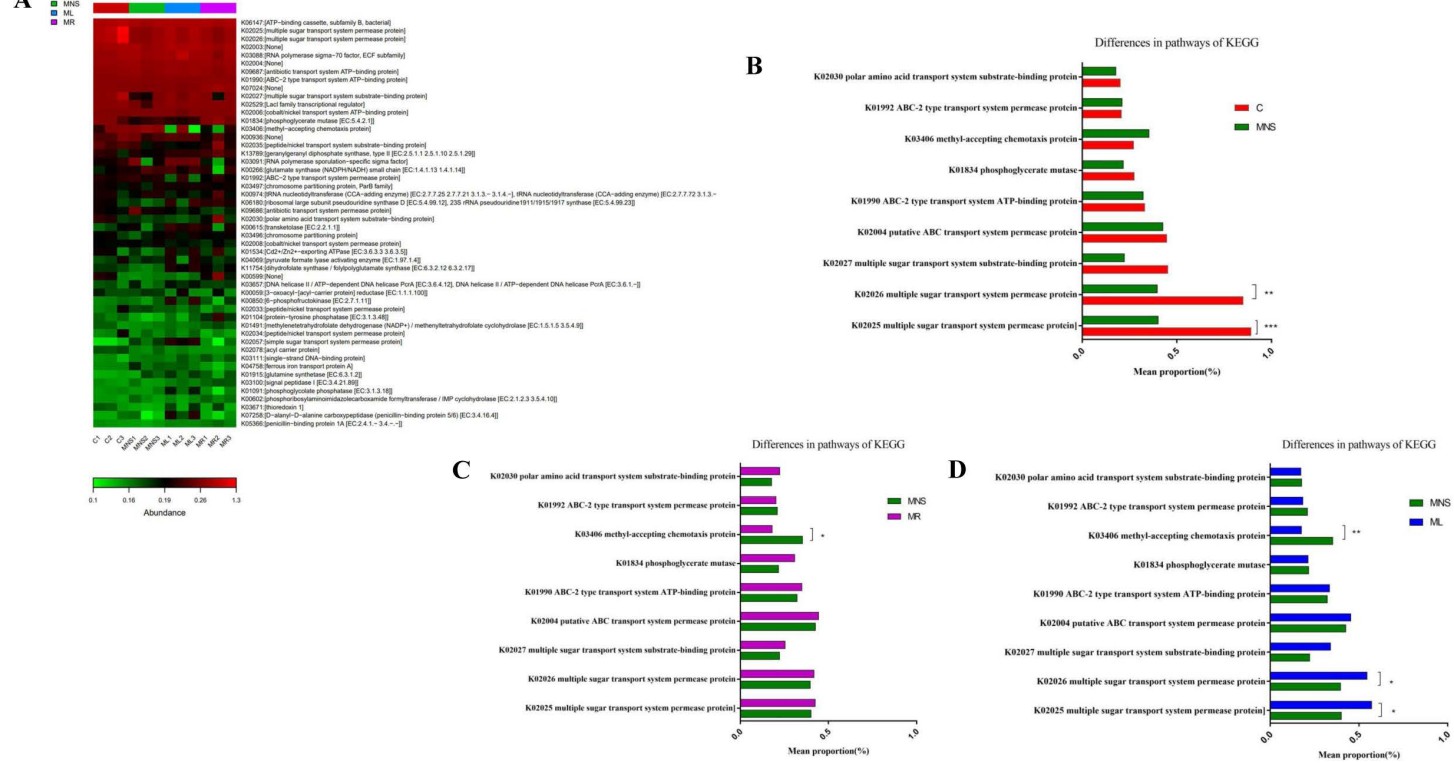

**Fig 7. Differences in microbial functions among groups in pathways of KEGG. (A)** KEGG functional abundance heatmap, drawn using a functional abundance matrix, where each column represents a sample, rows represent functions, and color blocks represent functional abundance values. The redder the color, the higher the abundance, while the greener the color, the lower the abundance. The functional abundance map indicates differences in metabolic pathways among the four groups. **(B)** Comparison of differences in metabolic functions between C and MNS group in pathways of KEGG. **(C)** Comparison of differences in metabolic functions between MNS and MR group in pathways of KEGG. **(D)** Comparison of differences in metabolic functions between MNS and ML group in pathways of KEGG. *$p < 0.05$, as determined by Mann-Whitney U test between two groups.

permease protein, $p = 0.0111$; and K02026 multiple sugar transport system permease protein, $p = 0.0344$), but the pathway of K03406 methyl-accepting chemotaxis protein ($p = 0.0084$) showed a downward trend (Fig 7D).

## 4 Discussion

The current landscape of therapeutic interventions for MHE primarily revolves around the modulation of gut microbiota, with lactulose and rifaximin being the most prominent agents [18–20]. Despite their widespread use, the exact mechanisms by which these treatments exert their effects remain inadequately understood. The primary objective of this study was to investigate the effects of rifaximin and lactulose on the gut-liver-brain axis in a rat model of MHE. Our key findings demonstrated that both rifaximin and lactulose significantly decreased serum and cerebrospinal fluid ammonia levels, improved cognitive deficits in MHE rats, and reduced systemic inflammation, specifically by lowering portal LPS levels and serum TNF-α and IL-1β. While liver function was not improved by these treatments, rifaximin enhanced intestinal barrier integrity by increasing occludin expression in the gut.

Jiang et al. confirmed that BAEP I latency is a dependable diagnostic marker for MHE in rats [21,27], and our study also utilized BAEP I latency, proving more precise than behavioral observations in diagnosing MHE. In our study, all MHE rats displaying latencies greater than 1.45ms, then treatment with rifaximin and lactulose significantly shortened these latencies. Using the MWM, we observed that MHE rats took longer to find the platform, but this delay was reduced with

rifaximin and lactulose treatment, aligning with clinical results of lactulose in MHE patients [28]. Furthermore, our findings showed that rifaximin and lactulose did not enhance liver function measures, which is consistent with prior studies [27,29]. However, some clinical studies have shown that rifaximin can improve liver function in patients with HE [20,30]. These differences may be due to differences in species, models, or HE severity. We observed that treatment with rifaximin and lactulose significantly reduced serum and cerebrospinal fluid ammonia levels in MHE rats, likely through a combination of decreased production and enhanced excretion. Lactulose lowers intestinal pH, converting ammonia into the less absorbable ammonium ion and facilitating its excretion [30,31]. Rifaximin may reduce ammonia generation by inhibiting urease-producing bacteria, although conflicting evidence regarding its effect on gut microbiota composition leaves its precise mechanism uncertain [22,23,30]. These unresolved pathways warrant further investigation.

Cirrhosis is associated with systemic inflammation, which promotes the progression of MHE and worsens neurocognitive dysfunction [32,33]. In our study, both rifaximin and lactulose significantly reduced portal LPS and serum TNF-α and IL-1β levels, supporting their anti-inflammatory effects. Previous studies have shown rifaximin alleviates endotoxin-induced inflammation [22,34,35], while lactulose reduces endotoxin levels and hepatic inflammation in rats [27,36]. Hematoxylin and eosin staining in our study confirmed that rifaximin and lactulose reduced liver cell disorganization and inflammatory infiltration in MHE rats. Rifaximin also significantly lowered TLR4 expression, consistent with findings that it inhibits the LPS/TLR4 pathway in liver fibrosis models [34]. Although the exact mechanisms by which rifaximin and lactulose reduce serum endotoxin levels in MHE have not been fully established, we propose potential mechanisms involving alterations in gut microbiota metabolism or enhancement of intestinal barrier function. Consequently, we further investigated the intestinal barrier and microbiome across the four study groups.

Our study found that rifaximin, but not lactulose, enhanced the intestinal barrier by upregulating occludin, a key tight junction protein. This is in line with observations in prior animal studies where rifaximin elevated occludin expression, potentially fortifying the gut barrier [37,38]. Specifically, Kuti et al. noted higher occludin mRNA levels in colonic tissues post-rifaximin treatment, indicating a possible mechanism for the protective effect of rifaximin on the gut [39]. In a model using humanized mice, rifaximin mitigated inflammation despite no change in mucosal permeability [22], mirroring its benefits in stressed mice [39]. Conversely, lactulose, while not affecting tight junction proteins, may improve intestinal transit and reduce permeability, offering an alternative pathway to hinder bacterial and endotoxin translocations [31]. Our exploration of these mechanisms suggests that both agents, through distinct actions, could significantly impede LPS from entering the liver, thus curbing the hepatic LPS/TLR4 pathway and providing therapeutic benefits.

In MHE, gut dysbiosis is characterized by a shift in bacterial populations, with a reduction in beneficial species like *Clostridiales XIV*, *Lachnospiraceae*, and *Ruminococcaceae*, and an increase in potentially harmful gram-negative bacteria such as *Enterobacteriaceae*, and *Staphylococcaceae* [13,40–42]. Our PAC and PCoA analyses confirmed these trends without a significant loss of diversity. Notably, MHE rats showed a slight decrease in *Clostridiales XIV* and *Bacteroides*, and an increase in *Saccharibacteria*, with a more pronounced decrease in *Lactobacillus*. Treatment with rifaximin and lactulose did not alter the overall diversity of the gut microbiota in MHE patients, yet it led to significant changes in specific bacterial groups. Rifaximin increased *Lactobacillus* abundance, while the combination treatment reduced *Saccharibacteria* to levels comparable to healthy controls. These findings align with previous studies showing minimal effects of rifaximin on the overall fecal microbiome composition [17,20,23]. While rifaximin has been shown to suppress deoxycholic acid production and reduce endotoxemia in cirrhotic patients, it does not significantly alter microbial diversity [17]. Similar studies also reported only modest changes in specific bacterial genera with rifaximin or lactulose treatment [20,41]. These findings suggest that both agents may reduce LPS and ammonia through mechanisms independent of major shifts in gut microbiota composition. Therefore, we used PICRUSt to predict the metabolic pathways influenced by gut microbiota.

To further investigate these mechanisms, we employed PICRUSt to predict the metabolic pathways associated with the gut microbiota. Our analysis indicated a decrease in glycometabolism following treatment with rifaximin and lactulose. Prior research has underscored that rifaximin is linked to increased concentrations of both saturated and

unsaturated fatty acids, as well as carbohydrate metabolism intermediates, which may have implications for brain function [20,43,44]. These insights lend credence to the hypothesis that the predominant action of rifaximin and lactulose is to modulate bacterial metabolic functions instead of primarily reducing bacterial abundance. However, while our PICRUSt-based predictions suggest potential modulation of host metabolic pathway, the current study does not directly measure functional metabolites or validate the activity of these pathways. For instance, the predicted upregulation of glycometabolism in rifaximin-treated rats could imply enhanced microbial saccharolytic fermentation, potentially increasing short-chain fatty acid (SCFA) production-a critical link between gut microbiota and host energy metabolism [45]. Similarly, the downregulation of methyl-accepting chemotaxis proteins may reflect reduced bacterial motility and pathogenicit [46]. To substantiate these hypotheses, future studies should integrate targeted metabolomics with transcriptomic profiling of hepatic pathways such as peroxisome proliferator-activated receptor gamma, which is central to glucose and lipid homeostasis [47].

It is crucial to recognize the limitations present in our study. First, the scope of our investigation into the intestinal barrier was constrained by the limited availability of detection methods. This limitation hindered us from conducting a comprehensive assessment of the intestinal barrier integrity. Additionally, our study did not delve into the specific metabolites produced by the gut microbiota. Consequently, the precise impact of rifaximin and lactulose on these metabolites remains unexplored. Finally, while we propose that microbiota-mediated metabolic reprogramming contributes to the therapeutic effects of rifaximin and lactulose, the lack of direct pathway validation limits mechanistic certainty. In future studies, we plan to incorporate targeted metabolomic and transcriptomic analyses, including the quantification of key microbial-derived metabolites and host signaling pathways, to provide a more mechanistic understanding of how rifaximin and lactulose exert their therapeutic effects.

## 5 Conclusion

In summary, our research shows that rifaximin and lactulose can improve cognitive function in a rat model of MHE by affecting gut microbiota metabolism and strengthening the intestinal barrier. These changes help lower ammonia levels, reduce LPS transfer to the liver, and decrease inflammation in both the liver and the body. This study enhances our understanding of MHE and highlights gut microbiota-targeted treatments as a promising therapeutic approach.

## Supporting information

**S1 Fig. Rifaximin and lactulose on hepatic function in MHE rats.** (A) Serum ALT, (B) AST, (C) ALB, and (D) TBIL levels were assessed as indicators of liver injury.
(PNG)

**S2 Fig. Differential genera features based on DESeq2 analysis.** (A) Changes in relative abundance of the gut microbiota between C and MNS group. (B) Changes in relative abundance of the gut microbiota between MNS and MR group. (C) Changes in relative abundance of the gut microbiota between MNS and ML group.
(PNG)

**S3 Fig. Relative abundance at the genus level for *Akkermansia*, *Bacteroides*, *Bifidobacterium*, *Clostridium_XlVa*, *Faecalibacterium*, *Lactobacillus*, *Prevotella*, *Ruminococcus*, *Saccharibacteria*, *Streptococcus*, and *Veillonella*.**
(PNG)

**S4 Fig. The expression of Claudin-1 and ZO-1 in intestinal wall.**
(PNG)

**S5 Fig. The expression of GABARAP in rat brain tissue by IHC.**
(PNG)

**S1 Table. MHE diagnosis and escape latency after model building.**
(DOCX)

**S1 File. Original images of WB.**
(PDF)

**S2 File. Original datas.**
(DOCX)

**S3 File. Supplementary experimental procedures.**
(DOCX)

## Author contributions

**Conceptualization:** Xueyan Lin, Shiyun Lu, Rongrong Chen.

**Data curation:** Xueyan Lin, Zhengchao Zhang.

**Investigation:** Shiyun Lu, Rongrong Chen.

**Methodology:** Xueyan Lin, Zhengchao Zhang, Rongrong Chen.

**Project administration:** Yi Lin.

**Validation:** Yi Lin.

**Writing – original draft:** Xueyan Lin, Zhengchao Zhang.

**Writing – review & editing:** Shiyun Lu, Rongrong Chen.

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
