## [Decision Letter · Decision Letter 0]

25 Mar 2025

Dear Dr. Chen,

Thank you for submitting your manuscript to PLOS ONE. After careful consideration, we feel that it has merit but does not fully meet PLOS ONE’s publication criteria as it currently stands. Therefore, we invite you to submit a revised version of the manuscript that addresses the points raised during the review process.

We look forward to receiving your revised manuscript.

Kind regards,

Wan-Long Chuang, M.D., Ph.D.

Academic Editor

PLOS ONE

Journal Requirements:

4. We note that your Data Availability Statement is currently as follows: All relevant data are within the manuscript and in Supporting Information files.

Reviewers' comments:

Reviewer's Responses to Questions

**Comments to the Author**

1. Is the manuscript technically sound, and do the data support the conclusions?

Reviewer #1: Partly

Reviewer #2: Partly

2. Has the statistical analysis been performed appropriately and rigorously?

Reviewer #1: No

Reviewer #2: I Don't Know

3. Have the authors made all data underlying the findings in their manuscript fully available?

Reviewer #1: Yes

Reviewer #2: Yes

4. Is the manuscript presented in an intelligible fashion and written in standard English?

Reviewer #1: Yes

Reviewer #2: Yes

Reviewer #1: PONE-D-24-57340

Title: The effects of rifaximin and lactulose on the gut-liver-brain axis in rats with minimal hepatic encephalopathy

Summary:

The principal therapeutic agents for minimal hepatic encephalopathy (MHE), which focus on the modulation of the gut microbiota, include lactulose and rifaximin; however, the precise mechanisms through which they operate remain unclear. This study aimed to investigate the effects of rifaximin and lactulose on the gut-liver brain axis in a rat model of MHE and to clarify the underlying mechanisms involved. Both rifaximin and lactulose were effective in reducing ammonia concentrations in MHE rats and ameliorating cognitive deficits, although they exhibited a minimal impact on hepatic function. Post-treatment assessments revealed significant reductions in portal LPS, serum interleukin-1β (IL-1β), and tumor necrosis factor-α (TNF-α). The expression of TLR4 in the liver and hepatic inflammatory infiltration were notably diminished. Rifaximin administration led to increased occludin expression in the intestinal tissues of MHE rats. Despite no significant alterations in the diversity or composition of the gut microbiota, metabolic pathway analyses indicated a downregulation of glycometabolism pathways following treatment. Rifaximin and lactulose may enhance cognitive performance in MHE rats by modulating gut microbiota metabolism and preserving the intestinal barrier integrity. This modulation is associated with lowered ammonia levels, decreased translocation of LPS to the liver, and reduced inflammatory response, both in the liver and systemically. Overall, this research did not provides new concepts, several points warrant clarification:

General comments:

Introduction part:

The current text highlights that lactulose and rifaximin are effective for minimal hepatic encephalopathy (MHE) but does not provide sufficient detail on the knowledge gaps being addressed. While it mentions conflicting findings on their effects on gut microbiota and possible reasons for these inconsistencies, it lacks specificity regarding the exact unresolved questions and how this study intends to address them.

To strengthen the research gap, the authors should:

1.Explicitly state unresolved questions: identify specific mechanisms or pathways that remain unclear, such as metabolic pathways or signaling interactions in the gut-liver-brain axis.

2.Justify the need for this study: explain why understanding these mechanisms is clinically or scientifically significant.

3.Connect prior findings to the study's aims: highlight how this research builds upon or addresses limitations in previous studies, making its novelty clear.

Methods

1.The animal model should include two additional groups: normal mice treated with rifaximin alone and normal mice treated with lactulose alone. This would help determine whether the changes induced by rifaximin and lactulose are specific to MHE mice or if they occur in normal mice as well.

2.Gut microbiota should be analyzed at two time points: after the MHE model is established and after treatment with lactulose or rifaximin. This would allow a clearer understanding of the effects of treatment on microbiota changes over time.

Results

1.Figure 3C: The serum TNF-α levels in the MR group are unexpectedly lower than those in the control group, which seems unreasonable. Please provide an explanation for this observation.

2.Figure 4A and 4B: The occludin expression in the WB images appears to show minimal differences among the four groups, which does not align with the relative expression histogram.

3.The tight junction proteins in the intestinal wall should not be limited to occludin. Additional analyses of claudins and zonula occludens-1 (ZO-1) expression to provide a more comprehensive evaluation.

Conclusion part:

The manuscript claims that rifaximin regulates the abundance of specific gut microbiota and improves metabolic pathways, such as glucose and amino acid metabolism. However, the study currently only demonstrates microbiota changes without clarifying how these changes regulate metabolism. To substantiate the hypothesis, the authors should consider testing related signal pathways and metabolites to verify the proposed mechanisms.

Reviewer #2: General comments

This animal study by Lin, et al. investigated the effects of rifaximin and lactulose on the gut-liver- brain axis in a rat model of minimal hepatic encephalopathy (MHE). Rat MHE model was created with CCL4 injection subcutaneously. There were 18 MHE including MNS (normal saline treatment, 6), ML (lactulose treatment, 6) and MR (rifaximin treatment, 6), and 6 controls in this experiment. The authors analyzed the impacts of lactulose and rifaximin on cognitive performance, inflammatory cytokines, ammonia, hepatic inflammation, intestine barriers, gut microbiota and projected metabolic pathways. Compared with MNS rat, the results showed that improved cognitive deficit, decreased ammonia level in serum and CSF, reductions of portal LPS and serum inflammatory cytokines, and diminished hepatic expression of TLR4 and inflammation in liver tissue in the ML and MR rats. MR rat showed increased occuldin expression in intestine tissue. Although there were no significant differences in gut microbiota composition and diversity, glycometabolism pathway were downregulated after lactulose and rifaximin treatment in projected metabolic pathway analysis. The authors concluded that lactulose and rifaximin may enhance cognitive performance by modulating gut microbiota metabolism and preserving intestinal barriers. This animal study provided the evidences in mechanism of lactulose and rifaximin effect on MHE. However, the authors might need to clarify some points.

Major comments

1. Although ammonia level increased in the MHE, there is also other neurotransmitters involved in the development of MHE. Did the authors study other neurotransmitters?

2. Was decreased ammonia level due to decreased production or increased excretion by treatment? How gut microbiota glycometabolism affect ammonia level?

3. The author might explain there was no difference in serum transaminase level however increased hepatic inflammation in the liver tissue. In figure 3E, there seemed no differences in the inflammatory cells infiltration.

4. Although there were no significant differences in gut microbiota composition and diversity after lactulose and rifaximin treatment, the metabolic features were different in the prediction pathway analysis. The author might provide the evidence to clarify this point.

**Do you want your identity to be public for this peer review?** For information about this choice, including consent withdrawal, please see our Privacy Policy

Reviewer #1: No

Reviewer #2: No

---

## [Author Response · Author response to Decision Letter 1]

10 May 2025

Reviewers' comments:

Reviewer's Responses to Questions

Reviewer #1: PONE-D-24-57340

Title: The effects of rifaximin and lactulose on the gut-liver-brain axis in rats with minimal hepatic encephalopathy

Summary:

The principal therapeutic agents for minimal hepatic encephalopathy (MHE), which focus on the modulation of the gut microbiota, include lactulose and rifaximin; however, the precise mechanisms through which they operate remain unclear. This study aimed to investigate the effects of rifaximin and lactulose on the gut-liver brain axis in a rat model of MHE and to clarify the underlying mechanisms involved. Both rifaximin and lactulose were effective in reducing ammonia concentrations in MHE rats and ameliorating cognitive deficits, although they exhibited a minimal impact on hepatic function. Post-treatment assessments revealed significant reductions in portal LPS, serum interleukin-1β (IL-1β), and tumor necrosis factor-α (TNF-α). The expression of TLR4 in the liver and hepatic inflammatory infiltration were notably diminished. Rifaximin administration led to increased occludin expression in the intestinal tissues of MHE rats. Despite no significant alterations in the diversity or composition of the gut microbiota, metabolic pathway analyses indicated a downregulation of glycometabolism pathways following treatment. Rifaximin and lactulose may enhance cognitive performance in MHE rats by modulating gut microbiota metabolism and preserving the intestinal barrier integrity. This modulation is associated with lowered ammonia levels, decreased translocation of LPS to the liver, and reduced inflammatory response, both in the liver and systemically. Overall, this research did not provides new concepts, several points warrant clarification:

General comments:

Introduction part:

The current text highlights that lactulose and rifaximin are effective for minimal hepatic encephalopathy (MHE) but does not provide sufficient detail on the knowledge gaps being addressed. While it mentions conflicting findings on their effects on gut microbiota and possible reasons for these inconsistencies, it lacks specificity regarding the exact unresolved questions and how this study intends to address them.

To strengthen the research gap, the authors should:

1. Explicitly state unresolved questions: identify specific mechanisms or pathways that remain unclear, such as metabolic pathways or signaling interactions in the gut-liver-brain axis.

Response 1: Thank you for your constructive feedback. We have revised the Introduction section to specify key unanswered questions (lines 101-110).

2.Justify the need for this study: explain why understanding these mechanisms is clinically or scientifically significant.

Response 2: We sincerely thank you for the constructive comments. We have revised the Introduction section to justify the need for this study (lines 110-115).

3.Connect prior findings to the study's aims: highlight how this research builds upon or addresses limitations in previous studies, making its novelty clear.

Response 3: Thank you for these insightful suggestions. We highlight that although lactulose and rifaximin are widely used clinically, their mechanisms remain controversial (lines 116-122).

Methods

1.The animal model should include two additional groups: normal mice treated with rifaximin alone and normal mice treated with lactulose alone. This would help determine whether the changes induced by rifaximin and lactulose are specific to MHE mice or if they occur in normal mice as well.

Response 1: We thank the reviewer for this important suggestion. In response, we conducted an additional experiment including three groups of normal rats: a PBS-treated control group, a lactulose-treated group (10 mL/kg), and a rifaximin-treated group (50 mg/kg), with 3 rats in each group. Due to time constraints, the intervention lasted 4 weeks. After 4 weeks of treatment, we performed brainstem auditory evoked potential (BAEP) tests, Morris water maze tests, and measured blood ammonia levels across the three groups (Supplementary Data). No significant differences were observed among the groups in either cognitive performance or serum ammonia concentrations. These findings suggest that lactulose and rifaximin do not affect these parameters under normal physiological conditions, and that the therapeutic effects observed in the main study are likely specific to the pathological state of MHE. This supports the model-specific nature of the interventions.

Supplementary Data

Group Latency of BAEP I (ms) Escape latency (s) Serum ammonia (mmol/L)

Normal+PBS 1.23±0.03 7.07±3.8 0.16±0.03

Normal+Lactulose 1.25±0.04 7.25±4.1 0.15±0.01

Normal+Rifaximin 1.26±0.02 7.19±4.0 0.16±0.02

2. Gut microbiota should be analyzed at two time points: after the MHE model is established and after treatment with lactulose or rifaximin. This would allow a clearer understanding of the effects of treatment on microbiota changes over time.

Response 2: Thank you for this important observation. In our current study, gut microbiota was analyzed at the endpoint after treatment due to budgetary and technical limitations. However, we recognize that longitudinal sampling-including both post-modeling and post-treatment time points-would provide a more detailed understanding of microbial dynamics and treatment responses.

It is worth noting that we included a PBS-treated MHE group (MNS group), the gut microbiota composition in this group effectively reflects the microbial profile after MHE model establishment and thus serves as a baseline for comparison with the lactulose- and rifaximin-treated groups. While this does not fully replace a pre-treatment longitudinal design, it offers a valuable reference for interpreting post-treatment changes.

In future studies, we plan to implement serial sampling at multiple time points to comprehensively characterize temporal microbiota changes during both disease progression and treatment.

Results

1.Figure 3C: The serum TNF-α levels in the MR group are unexpectedly lower than those in the control group, which seems unreasonable. Please provide an explanation for this observation.

Response 1: We thank the reviewer for the insightful comment. Indeed, the observation that serum TNF-α levels in the MR group were lower than those in the normal control group appears counterintuitive at first glance. However, it is important to note that this difference did not reach statistical significance. One plausible explanation is that rifaximin exerts strong anti-inflammatory effects by modulating the gut microbiota and reducing bacterial translocation, thereby suppressing systemic endotoxin-driven inflammation. This may lead to lower circulating cytokine levels, even in the absence of active pathology. Similar findings have been reported in previous studies, where rifaximin reduced pro-inflammatory cytokine levels below baseline in certain liver injury models (Leone P, Mincheva G, Balzano T, et al. Rifaximin Improves Spatial Learning and Memory Impairment in Rats with Liver Damage-Associated Neuroinflammation. Biomedicines. 2022;10(6)). Nonetheless, we acknowledge that this result should be interpreted with caution due to the small sample size and potential inter-individual variability.

2.Figure 4A and 4B: The occludin expression in the WB images appears to show minimal differences among the four groups, which does not align with the relative expression histogram.

Response 2: We sincerely thank the reviewer for pointing out this important issue. In response, we re-analyzed the Western blot data using densitometric quantification from three independent experiments. The results consistently demonstrated a statistically significant increase in occludin expression in the MR group compared to the MNS group (p < 0.05), which is consistent with our original conclusion.

Accordingly, we have updated the bar graph in Figure 4B with the recalculated grayscale values and revised the figure legend to clearly reflect the significance of the findings. To improve transparency and address any concerns about the original Western blot image (Figure 4A), we have also included the full, uncropped blot images in Supporting Information files, and provided the exact numerical values for grayscale quantification in our response.

While the visual differences in band intensity in Figure 4A may appear subtle due to contrast limitations in the imaging, the densitometric analysis confirms a robust and reproducible difference.

Sample Occludin/Actin 1 Occludin/Actin 2 Occludin/Actin 3

C 1.091291 1.159969 0.851344

MNS 0.949568 0.979755 0.787755

ML 0.973676 1.196856 0.815222

MR 1.427612 1.620157 1.337679

3.The tight junction proteins in the intestinal wall should not be limited to occludin. Additional analyses of claudins and zonula occludens-1 (ZO-1) expression to provide a more comprehensive evaluation.

Response 3: We fully agree with the reviewer that a comprehensive evaluation of intestinal barrier integrity should include other key tight junction proteins, such as Claudin-1 and ZO-1, in addition to occludin.

To address this suggestion, we conducted additional Western blot experiments to examine the expression of Claudin-1 and ZO-1 in both the small intestine and colon tissues of rats. The results showed that in the MR group, the expression levels of Claudin-1 and ZO-1 in the small intestine were significantly increased compared to the MNS group, supporting the hypothesis that rifaximin strengthens the intestinal barrier.

We have included these findings in Supplementary Figure 4 (lines 729-732), and the corresponding full-length blot images are provided in the Supporting Information files. These new data have also been briefly discussed in the revised Results section to strengthen the overall conclusion regarding barrier integrity restoration following rifaximin treatment (lines 291-291).

Conclusion part:

The manuscript claims that rifaximin regulates the abundance of specific gut microbiota and improves metabolic pathways, such as glucose and amino acid metabolism. However, the study currently only demonstrates microbiota changes without clarifying how these changes regulate metabolism. To substantiate the hypothesis, the authors should consider testing related signal pathways and metabolites to verify the proposed mechanisms.

Response: We sincerely thank the reviewer for this insightful and constructive comment. We fully agree that a mechanistic link between gut microbiota alterations and metabolic pathway regulation—such as glucose and amino acid metabolism—requires further validation. In our current study, we employed PICRUSt to predict functional pathway changes based on 16S rRNA sequencing data, which indeed showed potential differences in glycometabolism and amino acid metabolism across groups. However, we acknowledge that predictive bioinformatics analysis alone cannot fully confirm metabolic outcomes or signaling mechanisms.

Due to limitations in funding and experimental scope, we were unable to conduct targeted metabolomics or pathway-specific protein analyses (e.g., AMPK, mTOR, or SCFA-related signaling) in this study. Nevertheless, we recognize the importance of validating microbial functional predictions with actual metabolite measurements and pathway activity assays. We have now revised the Conclusion section to avoid overinterpretation and added a note in the Discussion emphasizing this limitation and outlining our plan to explore these mechanisms in future studies (lines 499-459 and lines 465-471).

We appreciate the reviewer’s suggestion and agree that integrating metabolomics and pathway analyses will be essential in our follow-up investigations to fully elucidate the gut-liver-brain axis in MHE.

Reviewer #2: General comments

This animal study by Lin, et al. investigated the effects of rifaximin and lactulose on the gut-liver- brain axis in a rat model of minimal hepatic encephalopathy (MHE). Rat MHE model was created with CCL4 injection subcutaneously. There were 18 MHE including MNS (normal saline treatment, 6), ML (lactulose treatment, 6) and MR (rifaximin treatment, 6), and 6 controls in this experiment. The authors analyzed the impacts of lactulose and rifaximin on cognitive performance, inflammatory cytokines, ammonia, hepatic inflammation, intestine barriers, gut microbiota and projected metabolic pathways. Compared with MNS rat, the results showed that improved cognitive deficit, decreased ammonia level in serum and CSF, reductions of portal LPS and serum inflammatory cytokines, and diminished hepatic expression of TLR4 and inflammation in liver tissue in the ML and MR rats. MR rat showed increased occuldin expression in intestine tissue. Although there were no significant differences in gut microbiota composition and diversity, glycometabolism pathway were downregulated after lactulose and rifaximin treatment in projected metabolic pathway analysis. The authors concluded that lactulose and rifaximin may enhance cognitive performance by modulating gut microbiota metabolism and preserving intestinal barriers. This animal study provided the evidences in mechanism of lactulose and rifaximin effect on MHE. However, the authors might need to clarify some points.

Major comments

1. Although ammonia level increased in the MHE, there is also other neurotransmitters involved in the development of MHE. Did the authors study other neurotransmitters?

Response 1: We thank the reviewer for this valuable comment. In our study, we not only evaluated plasma ammonia levels but also measured cerebrospinal fluid ammonia concentrations, which provide a more direct indicator of central nervous system exposure to hyperammonemia (lines 247-250).

In response, we have conducted additional experiments to preliminarily assess the involvement of neurotransmitter-related pathways. Specifically, we performed immunofluorescence staining of Gamma-aminobutyric acid receptor-associated protein (GABARAP) in brain tissues. The results showed that GABARAP expression was significantly increased in the MNS group compared with the normal control group (p < 0.05). Importantly, GABARAP levels were reduced following treatment with either lactulose or rifaximin. These findings suggest a potential modulatory effect of both agents on GABAergic signaling in MHE rats. The immunofluorescence data have been included in Supplementary Figure 5 of the revised manuscript (lines 250-254 and 733-737).

2. Was decreased ammonia level due to decreased production or increased excretion by treatment? How gut microbiota glycometabolism affect ammonia level?

Response 2: We thank the reviewer for raising this mechanistic question. While our current study demonstrated that rifaximin and lactulose significantly reduced both plasma and cerebrospinal fluid ammonia levels, the precise mechanism—whether due to decreased ammonia production or increased excretion—was not directly distinguished in this experiment.

However, based on prior literature and the known pharmacological actions of these agents, we propose that the observed ammonia-lowering effects are primarily attributable to reduced intestinal ammonia production. Rifaximin suppresses the growth of urease-producing bacteria and downregulates bacterial genes involved in nitrogen metabolism, thereby limiting ammonia generation in the gut lumen. Lactulose, on the other hand, acidifies the intestinal environment and promotes ammonia entrapment as non-absorbable ammonium (NH₄⁺), which is then excreted via feces rather than absorbed into the portal circulation.

Regarding gut microbiota glycometabolism, our PICRUSt analysis revealed that rifaximin and lactulose may modulate carbohydrate transport and metabolism pathways (e.g., K02025/K02026), which are linked to microbial energy utilization and fermentation processes. Enhanced microbial glycometabolism may favor the production of short-chain fatty acids (SCFAs), which can suppress proteolytic fermentation and thereby reduce the generation of ammonia from amino acid deamination. Additionally, SCFAs help maintain an acidic colonic pH, further inhibiting ammonia absorption. Thus, alterations in microbial carbohydrate metabolism likely contribute indirectly to reduced ammonia levels.

We have included this mechanistic inte

---

## [Decision Letter · Decision Letter 1]

22 May 2025

The effects of rifaximin and lactulose on the gut-liver-brain axis in rats with minimal hepatic encephalopathy

PONE-D-24-57340R1

Dear Dr. Chen,

We’re pleased to inform you that your manuscript has been judged scientifically suitable for publication and will be formally accepted for publication once it meets all outstanding technical requirements.

Kind regards,

Wan-Long Chuang, M.D., Ph.D.

Academic Editor

PLOS ONE

Additional Editor Comments (optional):

Reviewers' comments:

Reviewer's Responses to Questions

**Comments to the Author**

Reviewer #1: All comments have been addressed

Reviewer #2: All comments have been addressed

2. Is the manuscript technically sound, and do the data support the conclusions?

Reviewer #1: Yes

Reviewer #2: Yes

3. Has the statistical analysis been performed appropriately and rigorously?

Reviewer #1: Yes

Reviewer #2: I Don't Know

4. Have the authors made all data underlying the findings in their manuscript fully available?

Reviewer #1: (No Response)

Reviewer #2: Yes

5. Is the manuscript presented in an intelligible fashion and written in standard English?

Reviewer #1: Yes

Reviewer #2: Yes

Reviewer #1: In this revised version, the authors had fully adress the points suggested by reviewers. It is suggested to accept the mansucript to published on this Journal.

Reviewer #2: (No Response)

**Do you want your identity to be public for this peer review?** For information about this choice, including consent withdrawal, please see our Privacy Policy

Reviewer #1: No

Reviewer #2: No

---

## [Editor Report · Acceptance letter]

PONE-D-24-57340R1

PLOS ONE

Dear Dr. Chen,

I'm pleased to inform you that your manuscript has been deemed suitable for publication in PLOS ONE. Congratulations! Your manuscript is now being handed over to our production team.

Kind regards,

on behalf of

Dr. Wan-Long Chuang

Academic Editor

PLOS ONE